# Novel Technological Advances in Functional Connectomics in *C. elegans*

**DOI:** 10.3390/jdb7020008

**Published:** 2019-04-23

**Authors:** Elizabeth M. DiLoreto, Christopher D. Chute, Samantha Bryce, Jagan Srinivasan

**Affiliations:** Biology and Biotechnology Department, Worcester Polytechnic Institute, Worcester, MA 01605, USA; emdiloreto@wpi.edu (E.M.D.); cdchute@wpi.edu (C.D.C.); sbryce@wpi.edu (S.B.)

**Keywords:** connectomics, synapses, calcium imaging, optogenetics, sonogenetics, multisensory integration, sensory-motor integration

## Abstract

The complete structure and connectivity of the *Caenorhabditis elegans* nervous system (“mind of a worm”) was first published in 1986, representing a critical milestone in the field of connectomics. The reconstruction of the nervous system (connectome) at the level of synapses provided a unique perspective of understanding how behavior can be coded within the nervous system. The following decades have seen the development of technologies that help understand how neural activity patterns are connected to behavior and modulated by sensory input. Investigations on the developmental origins of the connectome highlight the importance of role of neuronal cell lineages in the final connectivity matrix of the nervous system. Computational modeling of neuronal dynamics not only helps reconstruct the biophysical properties of individual neurons but also allows for subsequent reconstruction of whole-organism neuronal network models. Hence, combining experimental datasets with theoretical modeling of neurons generates a better understanding of organismal behavior. This review discusses some recent technological advances used to analyze and perturb whole-organism neuronal function along with developments in computational modeling, which allows for interrogation of both local and global neural circuits, leading to different behaviors. Combining these approaches will shed light into how neural networks process sensory information to generate the appropriate behavioral output, providing a complete understanding of the worm nervous system.

## 1. Introduction

The field of connectomics attempts to link brain function with behavior by comprehensively mapping the anatomical links between all constituent neurons within different brain regions [1,2]. *Caenorhabditis elegans,* a microscopic roundworm, has served as a model for macroscopic research for over six decades (Figure 1). To date *C. elegans* remains the only organism to be fully mapped at the level of the nervous system [3,4,5]. The nervous system of both sexes; hermaphrodite (302 neurons) and male (385 neurons) have been completely mapped at the level of electron microscopy [4,6]. This has served as a prototype for analytical studies of larger scale connectome networks. However, can this complete mapping of synaptic connections in the brain (connectome) shape the understanding of the mechanistic basis of behavior? In other words, does structural connectivity define function within the nervous system? One mindset is that wiring diagrams can serve as a starting point for generating mechanistic hypotheses for the investigation of the neural basis of behavior. Therefore, knowledge of connectome structure enables the generation of testable hypotheses about the specific as well as general roles of individual neurons [7]. However, the physical connectivity patterns of the *C. elegans* nervous system, are unable to predict how the nervous system functions as a whole in order to enact behaviors [8,9,10]. Along with synapses, most nervous systems also contain gap junctions, which mediate fast, potentially bidirectional electrical coupling between cells [11,12]. In addition, extrasynaptic signaling between neurons within nervous system happens via monoamines and neuropeptides, occurring primarily outside the synaptic connectome [13]. These signaling systems act over both short and long ranges and are independent of synaptic connections, allowing them to shape behavioral responses to either the same or different stimuli [13,14]. Therefore, a complete characterization of a neuronal dynamics under different conditions is essential for generating a complete functional understanding of the nervous system. Since behavior is not a linear summation of sensory information, the same circuit can lead to different behavioral outputs. This could be a result of molecular events, stimuli concentration or physiological states—all of which are non-linear in nature—generating different behaviors via the same connectome. Hence, we define “functional connectomics” as the study of the relationship between a neuron’s function and its connections—both anatomical and extrasynaptic.

## 2. Functional Characterization of Neural Circuits: Using Connectome to Generate Hypotheses

*C. elegans* neurons can be divided into three functional “classes” of neurons: sensory neurons, motor neurons, and interneurons or premotor neurons [7,9]. The sensory neurons have dendrites that extend to the tip of the nose and terminate into diverse ciliated structures to detect stimuli from the environment. These neurons account for a third of the neurons with more connections being pre-synaptic than post-synaptic. Conversely, motor neurons, another third of all neurons, have more post-synaptic connections. The remaining neurons are considered to be premotor interneurons, with large numbers of both pre- and post-synaptic connections [10]. Understanding the connections alone between these neuron classes helps increase our understanding of how a signal is transduced, processed, to ultimately produce behavioral outputs, as seen in previous work on *C. elegans*’ navigation [15].

One of the first major findings of the worm connectome from the structural connectivity data was the characterization of the mechanosensory circuitry [16]. Using laser ablations, components of a mechanosensory circuit were identified, consisting of sensory neurons, premotor interneurons, and ventral cord motorneurons, responsible for escape behavior in response to body touch [16]. By testing each neuron’s function by laser microsurgery, [16,17], a set of premotor interneurons were identified that control the direction of locomotion; six neurons that promoted forward locomotion and four neurons promoting backward locomotion [16]. This work not only opened up the genetic and molecular studies of the *C. elegans* touch circuit, but also implicated these major premotor interneurons in several other behaviors [18,19,20,21]. This study was a landmark in understanding how structural changes within the connectome can impact function and signaling between neurons.

The compact nervous system of *C. elegans* allows for a single neuronal class to be involved in the sensation of diverse stimuli or elicitation of different behaviors [7,9,22]. From a functional connectomic perspective, this suggests that not all stimuli utilize the same pathways and connections. This suggests a non-linearity and complexity in the information processing of stimuli, for example, the polymodal, amphid, single-ciliated, nociceptive neuron, ASH, detects a myriad of different mechano, osmo, and chemo stimuli that result in aversive behaviors [23,24,25,26,27,28,29]. The diversity in neuronal circuitries may also be due to the intracellular machinery used within individual neurons. *C. elegans* are equipped with a large set of G protein subunits that exhibit overlapping expression, rendering particular intracellular pathways important in different behavioral circuits [30]. The nematodes genome codes for 21 Gα protein subunits, and 2 subunits each of both Gβ and Gγ proteins [30]. Of the 21 Gα subunits, 16 are expressed throughout the chemosensory neurons, and many overlap in their expression profiles [30]. For example, on its own, ASH expresses ten different Gα subunits, while a different amphid, single-ciliated, ASE, expresses only three [30].

## 3. Technologies Employed to Unravel the Functional Connectome

One of the earliest methods to monitor neuron function was patch clamp electrophysiology [31,32,33,34]. This technique lends itself to understanding functional connectomics as it monitors the flow of ions across neuronal membranes [33]. In *C. elegans* it was used to understand the role of graded potentials, in opposition to all or nothing action potentials observed in mammals [31,32,33,34]. This method is rather invasive, requiring fixed samples of individual neurons for testing [35].

### 3.1. Neuronal Imaging in C. elegans

Optical techniques are available in many experimental systems but are highly applied in *C. elegans* research as the nematodes are optically transparent and can be imaged while fully intact. In addition, a variety of Genetically Encoded Calcium Indicators (GECIs) are available targeting individual neuron/s of interest with specific promoters [36,37]. These GECIs can specifically target the neuronal cell body and/or distribute throughout the entire neuron. Fluorescent imaging with GECIs has achieved rapid progress in visualizing Ca^2+^ flux at the levels of cell populations [38], single cells [39], or even subcellular compartments [40]. Among available GECIs, Green fluorescent Calmodulin M13 fusion Protein (GCaMP) is one of the most successful and popular, due to its ability to convey Ca^2+^ levels with impressive signal-noise ratios (Figure 2A) [41,42,43].

Calcium imaging of neurons is widely employed in *C. elegans* neuroscience as it allows for the activity of a single or multiple neurons to be monitored over different time scales. Advances in microfabrication technology have permitted the construction of well-controllable microenvironments for monitoring neural function in *C. elegans*. One of the first microfluidic devices used to monitor neuronal calcium dynamics was termed the ‘olfactory chip’. This device was used to examine stimulus-response relationships in chemosensory neurons over a short temporal timescale [44]. Over the last decade, the applications of microfabrication techniques have exponentially increased in neuroscience with chips being designed for high-throughput and high resolution- based applications [44,45,46,47,48].

### 3.2. Whole-Brain Imaging in C. elegans

Measuring neural activity of a single neuron over a short time course is helpful in identifying neuronal dynamics upon stimuli exposure and characterizing circuits of activity [31,33,44,45,47,49,50,51]. However, it offers little insight into the processes at play during long-term behaviors. To execute motor commands during a particular behavior, more is occurring than cross-talk via the connection between neurons. Global-brain or whole-brain imaging enables characterization of behaviors with the integration of sensory neurons with motor neurons to be quantified [40,52]. While imaging a single neuron, sensory or premotor neurons are often the focus [45,53], whole-brain imaging enables elucidating on the role of multiple neurons within the connectome simultaneously. Studies in zebrafish have been accomplished on a global-brain scale with single-neuron resolution with specific regard to motor neurons [54,55].

The first whole-brain calcium imaging experiment in *C. elegans* was conducted in an immobilized setting, imaged neural activity across 100 neurons in a small channel. This technique relied on existing neural maps to match captured neural responses to individual neurons [4,38]. This study achieved single-neuron resolution across most of the brain and, when combined with the extensive existing knowledge of *C. elegans* neural anatomy, supported identification of most neurons. Using this technology, it is possible to track the calcium changes through circuits, generating a brain state phase plot (Figure 2B) [38]. This temporal experiment was key in identifying repeating patterns of stimulation similar to models of central pattern or rhythmic motor generation [38,56,57,58]. Tracking the whole-brain changes in calcium signals during this repetitive behavior allows for different classes of neurons as well as other movement variables, such as speed, to be incorporated into the understanding of this behavior [38]. While this information generated is sufficient to understand the culmination of a behavior, the neural information can also be parsed out to understand the signaling occurring in each part of the action, such as moving forward, slowing, reversing, or turning, on the scale of an individual neuron or across the brain [38]. This study suggested that high-level organization of behavior is encoded in the brain by globally distributed, continuous, and low-dimensional dynamics [38].

Recent developments have allowed researchers to develop two methods for imaging the neurons of *C. elegans* while roaming. These techniques rely on simultaneously recording several neurons expressing a calcium indicator using spinning disk confocal microscopy to ultimately produce volumetric imaging [59,60]. Both methods allow for the simultaneous recording of approximately 80 neurons, gathering and correlating information on body posture and location in a moving worm. One method expresses both the calcium indicator and another fluorescent protein, and also infers body posture and position via head ganglia orientation [59]. In contrast, the other method utilizes a custom software, which ensures that the head of the worm is always in the correct location of the microscope stage despite the worm’s movement, and actively records the body’s position [60]. The ability to generate multi-neuron data of freely roaming worms will be vital in moving towards a more thorough understanding of the functional connectome [61].

### 3.3. Optogenetics and the Worm Connectome

Optogenetics offers temporal control of activity of individual neurons utilizing light-controlled ion channels [62]. This unique optical control of neurons allows researchers to probe neural circuits and investigate neuronal function in a highly specific and controllable fashion [63,64,65,66,67,68,69]. *C. elegans* is a popular platform for probing the nervous system at length, with scales spanning from synapse to whole circuit [63,64,67]. The initial studies on optogenetic control of *C. elegans* neurons involved using whole-field illumination together with specific genetic mutations, involving activation of Channelrhodopsin (ChR2) in excitable motor neurons [68,69]. Specific neurons or muscles expressing ChR2 can be quickly and reversibly activated by light in both live and behaving animals [68,69]. Expressing a light-sensitive protein in specific neurons using specific promoters highlighted that functional neuronal circuits during behaviors can be elucidated in *C. elegans* (Figure 2C).

A drawback of whole-field illumination was lack of cellular specificity as the illumination occurred over a region of the worm’s body. Newer technologies have been generated wherein targeted illumination of multiple fluorophores expressing optically sensitive proteins, and extensively employed to observe behavior [67,70,71,72]. Using these illumination systems, it is now possible to track a freely moving *C. elegans* and spatiotemporally excite and/or inhibit specific nodes of neural networks to probe for function across several types of locomotory behaviors. In addition, this technology enables the use of combinations of optogenetic tools and fluorescent GECIs with high reproducibility and light intensity control. In addition to development of illumination systems, progress has been made in terms of development of variants of the optogenetic proteins that both depolarize and hyperpolarize neurons over longer temporal scales [64]. These highly light-sensitive optogenetic tools, are highly effective and display fast kinetics, allowing better investigation prolonged neuronal activity states in *C. elegans.*

### 3.4. Computational Strategies

*C. elegans* is a powerful biophysical system that can be understood at different levels such as sensory stimulation and motor output. Computational models serve as an excellent platform to implicate how dynamics of different neurons can affect network connectivity [73]. The development of neuroimaging tools and technologies enable us to record neural dynamics over a longer timescale, which help generating dynamic models of synaptic connectivity. Newer computational approaches help elucidate the dynamic connectome by comparing longer timescale studies containing higher-dimensions of data [73].

An interesting application of computational strategies is the characterization of sensory-motor integration [74]. Sensory-motor integration attempts to understand the neural pathways from stimuli input to motor-neuron driven behavioral responses. Given the connectome data, physical circuits can be identified, and then simulated to understand responses [74]. Modeling the dynamics of the neural network is possible by combining the known structural connectome data of *C. elegans* with a physiologically model of a neuron [74]. Models such as probabilistic graphical models (PGMs), use known circuits of repetitive behavior, (Figure 2D), to predict responses to novel situations [75,76].

A recent study has developed the “dynome” of the worm nervous system [77]. This predictive system relies on the simulation of neural dynamics, or temporal experiments on multiple neurons, to allow application of stimuli to neurons to examine network properties. Such a model enables the user to apply or modify stimuli to the network, observe the neural dynamics on various time and population scales, and allow for network structural changes. Changing these parameters allows for calculation of neural response patterns associated with different stimuli. The strength of this approach lies in the ability of removing neurons from a network and studying the dynamics of the circuit [77,78]. This unique computational approach is a first step in making any nervous system’s architecture being able to predict a behavior based on network properties.

## 4. Analysis of the Functional Connectome

The development of the technologies discussed above, allow us to investigate specific aspects of the connectome, such as sensation or processing of diverse multisensory stimuli. Sensory systems continuously receive complex types of environmental input, which are processed by the nervous system to identify relevant facts about the surroundings and internally represent that information to help the animal successfully navigate its environment. This complex neural function is further complicated when combined with the internal physiological state of the animal. In *Drosophila*, recent work on the multilevel multimodal convergence circuit, which relies on multisensory integration, was limited by the lack of connectomic data unveiling multisensory neuronal convergence [79].

Multisensory integration in *C. elegans* falls into two broad categories: co-exposure to two distinct stimuli, aversive and attractive, or exposure to one stimuli in conjunction with an environmental indicator, both of which can be tested via avoidance assays [80,81,82]. Furthermore, with only sixteen pairs of chemosensory neurons, neuronal ablation techniques such as the laser ablation method as well as genetic ablations have become effective tools for understanding the diverse roles of individual sensory neurons within the realm of the connectome [83,84].

### 4.1. Divergent Functions within a Single Neuronal Class

Modifications in functional connections downstream of sensory neurons can be achieved by differential cellular signaling mechanisms both within sensory neurons and between sensory and premotor interneurons. The nociceptive sensory neuron ASH detects various stimuli; chemical, mechanical, osmotic, all of which result in the same behavioral outcome: avoidance [25,28,29,85,86,87]. Studies have characterized differential use of both intra- and inter-signaling molecules by ASH sensory neuron leading to the same behavior (Figure 3A) [85]. For example, nose touch avoidance, requires expression of *itr-1* in ASH neurons and is not required for osmotic aversive responses [85]. Conversely, specific genes within ASH (such as *osm-10*) are specific for detecting osmotic stress, and not required for tactile response. This implies that in addition to expression of specific G protein-coupled subunits, distinct downstream effectors within the signaling pathway are specifically recruited to initiate functional connections. [85].

ASH has synaptic connections with AVA, AVB, and AVD, AVE, the forward and backward premotor interneurons [4]. Differential release of glutamate from ASH neurons may activate different types of glutamate receptors on these premotor interneurons mediating the nociceptive escape response (Figure 3A) [25,28,87]. The glutamate receptor, *glr-1,* is utilized primarily in nose touch avoidance, in downstream, premotor neurons (Figure 3A) [29,86,87]. Weak activation of ASH, elicited by nose touch, activates non-NMDA ionotropic glutamate receptor (iGluR) subunits GLR-1 (Figure 3A). Hyperosmolarity evokes higher levels of Ca^2+^ release, activating the NMDA ionotropic glutamate receptor NMR-1 along with GLR-1 [87]. Therefore, differential activation of inter/intra-signaling pathways leads to specific downstream neurons establishing different functional connections within the structural connectome.

While certain intracellular components and synaptic connections are vital in some behaviors, they may be irrelevant in other behavioral circuits. One example of this is the amphid sensory neuron, ADL, and its involvement in the response to ascaroside #3 (ascr#3) (Figure 3B). Ascarosides (ascr) are small-molecule signals which serve diverse functions in inter-organismal chemical signaling [88,89,90,91]. Ascr#3 is a small-molecule pheromone that causes different behavioral responses in males and hermaphrodites. Males are attracted by ascr#3, hermaphrodites are repelled by the cue [89,92,93]. Hermaphrodites avoid ascr#3 via ADL chemical synaptic transmission, presumably, to the backward command interneurons AVA and AVD [4,92]. ADL neuron’s avoidance to ascr#3 is regulated via the gap junction hub-and-spoke RMG circuit, whereas the interneuron RMG serves as a hub to modulate sensory neuron responses [92,93]. RMG, through the activity level of the neuropeptide receptor *npr-1*, and input from the sensory neuron ASK, can inhibit ADL triggered avoidance by altering gap junction properties [92,93]. Therefore, chemical synapses are involved in the avoidance to ascr#3, whereas gap junctions are necessary for modulating the response in an *npr-1* dependent manner to elicit aggregation or attraction (Figure 3B).

### 4.2. Developmental Connectomics: Rewiring of the Connectome during Larval Development

Throughout development, the nervous system undergoes drastic changes in neuron number and neural connectivity. From sensory systems to neuromuscular junctions, new cellular components expand neural circuits as they differentiate from progenitors [4,94,95]. These structural changes at each larval stage update sensorimotor responses and adapt to changing body plans. Though the adult wiring diagram has long been completed, no other developmental stage or other organism has been mapped with such synaptic resolution [4,96]. However, emerging technologies has allowed neurogenesis of the connectome to be followed throughout the development of *C. elegans*, utilizing the well-established electron miscopy data of the adult worm and embryonic cellular information [97]. 

From the first cell division to hatching, the *C. elegans* connectivity matrix [6,96] can be followed throughout the development of the *C. elegans* [97]. A fertilized *C. elegans* undergoes the first round of divisions to form two cells called AB and P1, named for their anterior and posterior locations, respectively [98]. The AB cell gives rise to the neurons [95]. The larval connectome is a dynamic structure with connections changing very quickly over a given period of development. For instance, the first neuronal cell RMEV, is born just shy of 5 hours post fertilization. This neuron has no connections to the four other neurons present, ADF and AWB right and left. However, ten minutes later, RMEV temporarily becomes hub of connection within the group of >30 neurons [97]. At the 5 hours mark, connectivity is more complex with neurons forming feedback loops among other large scale networking features [99]. After 10 hours, the embryo hatches, entering the L1 stage of development [100]. 

Neuronal cells migrate from their point of origin during development and rewiring occurs throughout the development of the worm [97]. At the L1 larval stage, the worm has 222 neurons [94], 22 of which are motor neurons [101,102]. Of the 80 neurons added into adulthood, the majority of these neurons belong to the motor class [94]. The 22 motor neurons of the L1 worm migrate to innervate the dorsal region in adulthood with the addition of 36 ventral motor neurons throughout the larval stages [102]. The point of genesis of a neuron in development does not have a great influence on its level of connectivity, with no clear connection between cells having greater connections in adulthood with neurons that have a large number of connections during the development of the worm [97]. However, two studies which investigated the origins of the *C. elegans* connectome in the embryo suggest that there are some relationships between neurons that are born early versus those that are born later in development [97,103]. Both studies used a complex network approach combining elements of the published connectome with the birth times and spatial locations of neurons. Using this approach, the researchers found that as the *C. elegans* embryo develops, a neural network emerges that is shaped by their ancestral developmental cell lineages and proximal relationships between these cells [97]. The growth of the network transitions from an accelerated to a constant increase in the number of synaptic connections as a function of the neuronal number [103]. These investigations highlight the fact that a full understanding of the interplay between anatomical, functional, and behavioral changes across development, requires dynamic and structural models of complete neural circuits at different stages of development. 

### 4.3. Sex Differences in the Functional Connectome

Interestingly, the sex of the animal can establish the synaptic connection and function of a neuron. One example of a sex-specific circuit change is in the sensation of ascr#3 (Figure 3B). Ascr#3 is also sensed by ADF, but only in males, and hermaphrodites which have been masculinized through expression of *fem-3,* a sex-determination protein which inhibits the sexual regulator gene, *tra-1* [104,105]. Neuronal activation of ADF by ascr#3 also requires *mab*-3, which is naturally inhibited in hermaphroditic animals [104]. As ADL activation in males, results in attraction, masculinized ADF in hermaphrodites inhibits the aversive response to ascr#3. This inhibition may be taking place via extrasynaptic connections, or through serotonin signaling on downstream neuronal target of ADL (Figure 3B). Sex can also result in different physical circuits, where synapses between certain neurons are only present in males, and pruned in hermaphrodites [106].

There are 294 neurons that are present in both the hermaphrodite and male worm [104,107,108,109,110,111,112]. These common neurons constitute a significant portion of the 302 hermaphrodite neurons [4] but the male has an additional 83 neurons primarily localized in the nose and tail [113]. In embryonic development, only two sets of sex specific cells develop in both sexes, HSN and CEM; the other cells develop throughout the larval stages [95]. In hermaphrodites, HSN cells are motor neurons involved in egg laying. The male-specific CEMs are involved with detecting hermaphrodite pheromones and help innervate cephalic sensilla [108,114]. Programed apoptosis eliminates the unnecessary category of neuron in each sex [115,116]. Complete differentiation doesn’t occur embryonically but in larval stages when it is important to complete sexually different neural circuits prior to their use in adulthood [94,113]. Sexually dimorphic neuronal connectivity comes about primarily in the L4 stage, when sexual maturity is reached [4,6,117,118]. At this stage of development, pruning occurs in particular neurons which later have an impact of sex specific behaviors especially those related to mate finding. Beyond physical changes, pruning of cells impacts sensory circuits leading to sex-specific reception of chemosensory information [104,119,120,121,122].

The importance in pruning of connections is prevalent in the PHB and AVA connection. In worms with this connection, hermaphrodites and young worms, there is an avoidance to noxious chemicals, e.g. SDS, [82,117] closely related to kairomone secreted their predator *Pristionchus pacificus* [123]. This connection is pruned in L4 males and they do not avoid noxious chemicals [82,117,118]. This difference in behavior is necessary to alter the way that males seek mates. Males must actively seek a mate to reproduce and may perform more ecologically dangerous behaviors to pass genetic information. Hermaphrodites do not need to do this and take action to preserve life. Etiological studies [124] focusing on this valence have shown that which changes in food availability effects this connection. When there is a lack of food as a juvenile L1, the male neuronal pruning is altered, affecting reproduction efficacy as the male does not maintain contact with the hermaphrodite [125]. This behavior is rescued with the addition of food prior to the L3 state [118]. 

Sex-specific circuits have been identified that govern the male response to sex pheromones, demonstrating the importance of fully mapping neural circuits in both hermaphrodites and males [122]. This highlights the necessity of investigating how specific connections underlying a behavioral circuit is regulated by sex of the organism, not merely the requisite neuron, in order to generate a more complete functional connectome.

### 4.4. Modulation of Neural Circuits

Behavioral circuits are dependent on the state of the animal. While receptor expression profiles and the sex of the animal are set variables, more flexible states—such as the physiological state of the animal—shape and modulate these circuits. Sensory networks are altered by neuromodulators (neurotransmitters and neuropeptides) in a context specific manner; over varying distances and timescales. The effect of these modulations varies based on site of release and local concentration as governed by release, degradation, and reuptake of neuromodulators [7,10].

The neurotransmitter serotonin (5-HT) has been shown to have a large role in behaviors related to foraging, egg laying, and locomotion, dependent on the presence or absence of food, as expression levels are correlated with being either fed or starved [126]. Interestingly, it was found that the site of release is important, able to generate opposing effects of 5-HT-mediated locomotion [127]. These findings highlight how a single neurotransmitter, within the same circuit, can give rise to different synaptic strengths and fine-tuned behavioral outputs. Moreover, the same stimulus does not necessarily utilize the same circuit at different concentrations [126]. Furthermore, the duration of stimulus detection is coded into neural circuits suggesting a role of temporal activity in shaping functional circuits. For instance, avoidance to copper, is a short-term behavioral state mediated by a cross-talk ASI and ASH inhibition circuit that fine tunes the behavioral response. ASH neurons respond quickly and robustly in comparison to a slower, weaker response by ASI, which inhibits further ASH activation [128].

Worms also exhibit long-term behavioral states for example, roaming and dwelling states in the presence of food alternate, and last for minutes at a time (Figure 4). This switch is achieved via two opposing neuromodulators: serotonergic signaling promotes dwelling, whereas the neuropeptide PDF-1, pigment dispersing factor, promotes the roaming state [129]. The neurons that produce and respond to each neuromodulator form a distributed circuit independent to the classical wiring diagram, with several essential neurons that express each molecule (Figure 4). Serotonergic signaling through *mod-1* initiates and extends dwelling states by inhibiting the neurons that promote roaming, whereas PDF signaling through *pdfr-1* initiates and extends roaming states (Figure 4). Despite the compact size of the *C. elegans* nervous system, the serotonin and PDF that regulate roaming and dwelling each have several important sources, and their receptors each act in several target neurons. Strikingly, this functional circuit defies classical circuit logic of sensory to motor organization: motor and interneurons modulate the activity of sensory neurons [129]. This largely extrasynaptic, long-term timescale circuit has many potential inputs that can bias signaling of one state over the other.

Together, functional connectomes often take shape in drastically different ways than wiring diagrams suggest, with particular synaptic importance being dictated by physiological states and timescales. Additionally, functional circuits do not work in isolation, the final behavioral output is a readout of the fine tuning of multiple functional circuits converging to create a functional connectome.

Just as the internal state modulates the response to a particular cue, the presence of multiple stimuli is integrated into larger networks. How integration of cues allows for the modulation of circuits can further be exemplified by looking at threat tolerance. Well-fed *C. elegans* do not cross a high osmotic barrier to chemotax the food-odor attractant, diacetyl: the risk is not worth reward. However, animals which are deprived of food will cross the same osmotic barrier, presumably weighing that the risk no longer outweighs the reward [81]. This modulation requires slow accumulation of tyramine—the expression of which increases during extended times of starvation, thereby desensitizing ASH to the osmotic stressor—and requires a few hours of starvation to reach a level of tyramine which allows for the switch to decide to cross the osmotic barrier [81].

The aforementioned examples showcase the complexity underlying functional circuits, as there seem to be multiple levels of neuronal processing acting in parallel to finely adjust how the animal responds, including specific intercellular machinery allows for rapid adjustment of neuronal responses, thereby affecting the output, and these modulations can also take place over long time scales—not merely minutes, but instead hours. Therefore, to truly decipher a functional circuit, many different time points and stimulus concentrations must be investigated, as at one concentration and time scale there is likely hidden information acting at a deeper level.

## 5. Discussion and Future Directions

Nervous systems are comprised of structurally interconnected neuronal networks and brain regions with complex connectivity patterns [78]. As mapping and recording techniques become increasingly capable of capturing neural structure and activity across widely distributed circuits and systems, there is a growing need for new analytical tools and modeling approaches interpret these rich sets of “big data”. *C. elegans*, with its well characterized physical connections, provides a strong platform for functional connectomic analysis and elucidation of connectivity patterns. Future functional connectome studies will require a strong push for rapid whole-brain imaging techniques that maintain a resolution which allows for detailed analysis of individual neurons. Currently, these techniques are not optimized in many organisms. In the case of *C. elegans,* using a technique to image a single neuron requires the animal to be moved into a separate testing environment. By this methodology, the worm is often reduced to a fixed space [44]. This idea of imaging in vivo yet removed from the natural environment rings to the same tune as MRI or other imaging techniques that require large pieces of equipment. On the other end of this spectrum, exists techniques that do not even require a microscope. These lens-free methods include an optofluidic microscope: *C. elegans* are still able to move around freely in solution while their behavior is monitored [130]. This technique goes further than a behavioral assay, as it obtains real time results from internal structures.

Whole-brain imaging studies suggest that a population coding mechanism allows for the smooth transitioning of network activity [38,59,60]. This allows the worm to switch between different programs (forward to backward or vice versa) during locomotor behaviors. Novel computational methods will need to be developed to verify in a quantitative manner that population-level features indeed encode behaviors. Moreover, whole-brain imaging in freely moving worms should reveal whether other possible population-level features have indeed behavioral correlates. Optogenetics is especially well suited to uncovering compartment-specific processes. Developing the optogenetic toolkit further to localize photosensitive proteins to specific subcellular locations with precise activation is an area of future research. These technologies will help decipher subcellular dynamics of sensory and interneurons.

In silico approaches that utilize the *C. elegans* connectome to model known behaviors and also predict novel outcomes to a known stimulus are currently being developed. One such open-source platform is the OpenWorm, with the aim of building a complete digital organism to simulate all features of *C. elegans*’ behavior [131,132]. Computational modeling using novel algorithms and superimposition of these models on the experimental data can provide insights into how network/s function after stimulus exposure [73,74,75]. However, developing models that can predict network function based on simulations is still an area that requires further study. Additionally, dynamics of multitudes of neurons during a certain behavior, allow for new approaches in modeling incorporating both the structural connectome data and layering it with the neurophysiological responses and interactions [77]. The ‘Dynome’ model depicts the dynamical systems overlaying the structural connectivity [77]. These models are more akin to the realistic nervous system and have amazing potential for revealing novel neural pathways and functionalities of the network [7,10,78].

There has also been a substantial amount of developments in non-invasive techniques for probing neural mechanisms. One technique utilizes ultrasound waves to stimulate neural circuits in worms and other excitatory cells [133]. This field of sonogenetics delivers ultrasound to manipulate the neural circuit through a variety of mediums. One method uses repeated exposure of low-pressure ultrasound with microbubbles, while *C. elegans* remain on agar plates [134]. Others turn to microfluidic chip devices to deliver a single, short pulse of ultrasound [135]. As of 2016, the use of sonogenetics has been approved by the FDA to treat essential tremors in humans. This high-intensity focused ultrasound uses the mechanisms of MRI to map structures, before ablating damaged structures exacerbating tremors, typically localized in the thalamus [136].

In conclusion, connectomics (both structural and functional) are likely to expand significantly in coming years. Several large-scale national and international projects and consortia directed at brain science are underway, including the Human Connectome Project and the BRAIN initiative in the U.S. as well as the Human Brain Project in the E.U. [137,138,139]. Given the rate of data generation, an interesting avenue is development of frameworks that can span different scales and neural systems, helping make sense of “big brain data” [140].

## Figures and Tables

**Figure 1 jdb-07-00008-f001:**
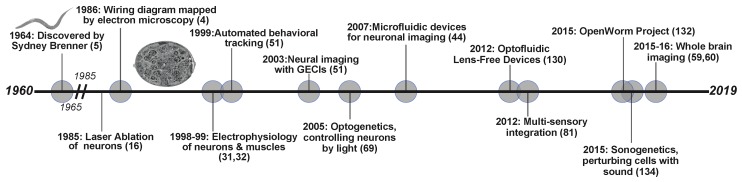
Timeline of neuroscience related discoveries in *C. elegans*. The past 60 years have resulted in development in technologies and breakthroughs in understanding the neural connections and the nervous system.

**Figure 2 jdb-07-00008-f002:**
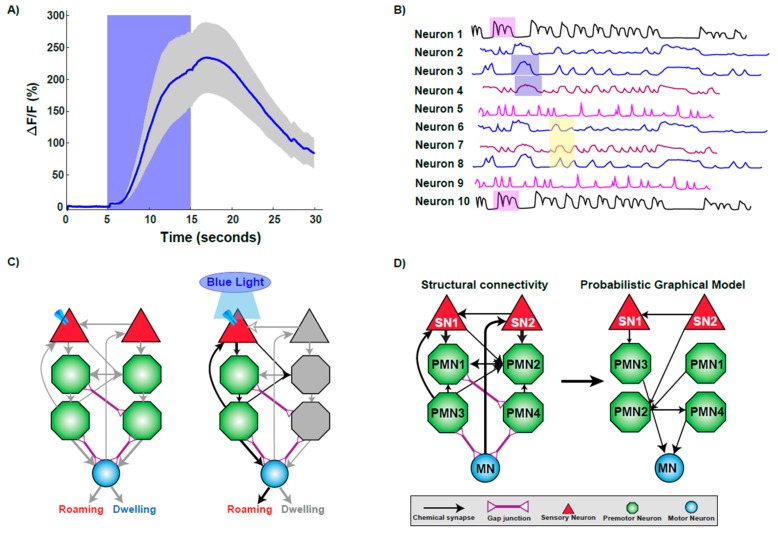
Technologies used to decipher the functional connectome. (**A**) Imaging calcium changes using GCaMP sensor. The trace plot shows the activation of a sensory neuron upon stimulus presentation by increase in calcium influx as measured by increase in fluorescence intensity. Increase in calcium is sustained while the worm experiences the stimulus, as fluorescence decreases upon stimulus removal. (**B**) A representative brain phase plot where neurons are activated in different phases (shaded regions) during exposure to a stimulus. (**C**) Optogenetic interrogation of the connectome. Expressing and activating Channelrhodopsin via blue light exposure in a particular sensory neuron activates a subset of downstream neurons, resulting in the elicitation of roaming behavior. (**D**) Computational modeling helps to unravel the functional connections within a structural framework of neurons.

**Figure 3 jdb-07-00008-f003:**
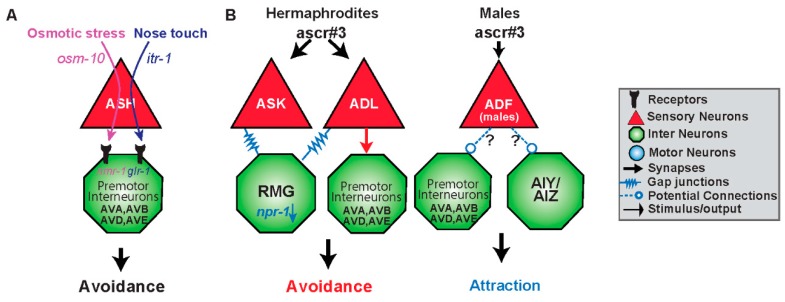
Functional circuits are created by differential use of neurons and distinct intra- and intercellular signaling pathways. (**A**) The sensory neuron ASH responds to two different stimuli resulting in an avoidance response via the premotor interneurons AVA, AVB, AVD and AVE. Osmotic stress is mediated via *osm-10* whereas nose touch utilizes *itr-1* within ASH. Downstream of ASH, osmotic stress targets the N-methyl-D-aspartate (NMDA)-type receptor *nmr-1*, whereas nose touch activates the glutamate receptor *glr-1* in the premotor interneurons [86,88]. (**B**) Gender and type of synapse govern behavioral output to the same stimulus. ADL senses the ascaroside, ascr#3, and in solitary hermaphrodites (high *npr-1*) results in avoidance via electrical synapses. However, hermaphroditic animals with low *npr-1* activity dampen or even reverse the valence of response to ascr#3 by the hub-and-spoke gap junction circuit, specifically ASK and RMG. Gender also shapes the response to ascr#3. Males are also able to detect ascr#3 via the masculine *mab-3* expressing state of ADF and are attracted to the compound. It is likely that ADF is opposing the ADL promoted avoidance response either via input to command interneurons or the first layer amphid interneurons it synapses with.

**Figure 4 jdb-07-00008-f004:**
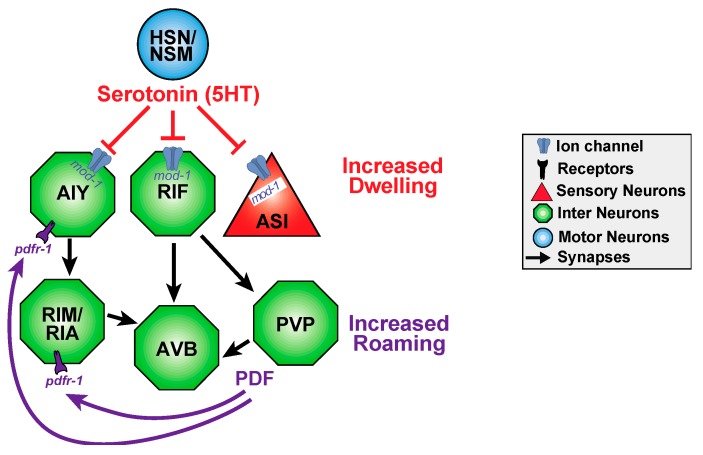
Functional circuits may be shaped by both neurotransmitters and neuropeptides and by short and long timescales. *C. elegans* display long-term behavioral dwelling or roaming states which are triggered by serotonin and PDF, respectively. PDF prolongs roaming and shortens dwelling states, whereas serotonin has reciprocal effects. PVP has been hypothesized to secrete PDF. Roaming and dwelling behavior seems to be modulated by a distributed circuit with the switch between dwelling and roaming is seemingly spontaneous. NSM neurons are implicated in feeding, and HSN neurons are implicated in egg laying. AIY and RIM neurons regulate reversal frequencies. ASI neurons are sensory neurons (triangles) that sense food, pheromones, to regulate dauer larva development. RIA interneurons regulate head curving during locomotion. AVB interneurons are forward command neurons in the motor circuit. Interestingly, functional connections can be extrasynaptic and defy sensory to motor circuit logic as HSN/NSM serotonin inhibits ASI in this behavior. Circuit adapted from Flavell et al. [129].

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
