# Peer review of "Novel Technological Advances in Functional Connectomics in C. elegans"

_jdb, 2019, doi:10.3390/jdb7020008_

Round 1

Reviewer 1 Report

The paper surveys methods and history of connectomics for C. elegans and highlights the importance of additional information included in molecular processes, extra-synaptic activity for detecting functional correlations and circuits within the nervous system.

The paper reads well and provides a coherent story of the developments (chronologically) in the field of C. elegans connectomics and their relevance to biological and behavioral consequences. 

Several areas of improvement:

* The title of the paper should be more focused and reflect the fact it is a review. For example: "The importance of novel technological advances for functional connectomics in C. elegans". Current title is too broad and suggests that the manuscript introduces novel methodologies for functional connectomics (which doesn't appear to be the case). 

* Precise definition of functional connectomics should be explained and elaborated. Current descriptions in the introduction (lines 22 - 36) are vague. E.g. line 35 "All behaviors result from summation of actions ..." is misleading since behavior is not necessarily summation (linear) operation. The same circuit can generate different actions (e.g. oscillatory activity, non oscillatory ) in different states. Alternative definition would be: various processes, e.g. molecular processes and stimuli, which are nonlinear and complex, generate different behaviors founded in the same connectome.

* Update of recent whole-brain imaging and modeling methods work is needed. Here are some links:

-Current state of the art achieves whole-brain imaging in freely moving worms.

Nguyen, Jeffrey P., et al. "Whole-brain calcium imaging with cellular resolution in freely behaving Caenorhabditis elegans." Proceedings of the National Academy of Sciences 113.8 (2016): E1074-E1081.

Venkatachalam, Vivek, et al. "Pan-neuronal imaging in roaming Caenorhabditis elegans." Proceedings of the National Academy of Sciences 113.8 (2016): E1082-E1088.

The discussion re. immobilized imaging should be revised accordingly.

-Progress have been made in modeling synaptic activity and showing activation of connectome:  

Kunert, James, Eli Shlizerman, and J. Nathan Kutz. "Low-dimensional functionality of complex network dynamics: Neurosensory integration in the Caenorhabditis elegans connectome." Physical Review E 89.5 (2014): 052805.

Kim, Jimin, William Leahy, and Eli Shlizerman. "Neural Interactome: Interactive Simulation of a Neuronal System." bioRxiv (2017): 209155.

Kunert-Graf, James M., et al. "Multistability and long-timescale transients encoded by network structure in a model of C. elegans connectome dynamics." Frontiers in computational neuroscience 11 (2017): 53.

-Computational work showed functional connectome could be derived from dynamics

Liu, Hexuan, Jimin Kim, and Eli Shlizerman. "Functional connectomics from neural dynamics: probabilistic graphical models for neuronal network of Caenorhabditis elegans." Phil. Trans. R. Soc. B 373.1758 (2018): 20170377.

Author Response

Reviewer 1

Comments and Suggestions for Authors

Reviewer Comment 1: The paper surveys methods and history of connectomics for C. elegans and highlights the importance of additional information included in molecular processes, extra-synaptic activity for detecting functional correlations and circuits within the nervous system.

The paper reads well and provides a coherent story of the developments (chronologically) in the field of C. elegans connectomics and their relevance to biological and behavioral consequences. 

Several areas of improvement:

* The title of the paper should be more focused and reflect the fact it is a review. For example: "The importance of novel technological advances for functional connectomics in C. elegans". Current title is too broad and suggests that the manuscript introduces novel methodologies for functional connectomics (which doesn't appear to be the case).

Author response:

We thank the reviewer for their suggestions for the title and agree that the title we used was very broad. We have now changed the title of the paper to “Novel Technological Advances in Functional Connectomics in C. elegans”.

Reviewer Comment 2:  Precise definition of functional connectomics should be explained and elaborated. Current descriptions in the introduction (lines 22 - 36) are vague. E.g. line 35 "All behaviors result from summation of actions ..." is misleading since behavior is not necessarily summation (linear) operation. The same circuit can generate different actions (e.g. oscillatory activity, non oscillatory ) in different states. Alternative definition would be: various processes, e.g. molecular processes and stimuli, which are nonlinear and complex, generate different behaviors founded in the same connectome.

Author response: 

We agree that the precise definition of functional connectomics was not elaborated in the original version of the manuscript. The revised version of the manuscript addresses this concern (lines 65-70). Line 68-70: “Hence, we define the “functional connectome” as the study of the relationship of a neuron’s function to its connections, both anatomical and extrasynaptic.”

Reviewer Comment 3: Update of recent whole-brain imaging and modeling methods work is needed. Here are some links:

-Current state of the art achieves whole-brain imaging in freely moving worms.

Nguyen, Jeffrey P., et al. "Whole-brain calcium imaging with cellular resolution in freely behaving Caenorhabditis elegans." Proceedings of the National Academy of Sciences 113.8 (2016): E1074-E1081.

Venkatachalam, Vivek, et al. "Pan-neuronal imaging in roaming Caenorhabditis elegans." Proceedings of the National Academy of Sciences 113.8 (2016): E1082-E1088.

The discussion re. immobilized imaging should be revised accordingly.

-Progress have been made in modeling synaptic activity and showing activation of connectome:  

Kunert, James, Eli Shlizerman, and J. Nathan Kutz. "Low-dimensional functionality of complex network dynamics: Neurosensory integration in the Caenorhabditis elegans connectome." Physical Review E 89.5 (2014): 052805.

Kim, Jimin, William Leahy, and Eli Shlizerman. "Neural Interactome: Interactive Simulation of a Neuronal System." bioRxiv (2017): 209155.

Kunert-Graf, James M., et al. "Multistability and long-timescale transients encoded by network structure in a model of C. elegans connectome dynamics." Frontiers in computational neuroscience 11 (2017): 53.

-Computational work showed functional connectome could be derived from dynamics

Liu, Hexuan, Jimin Kim, and Eli Shlizerman. "Functional connectomics from neural dynamics: probabilistic graphical models for neuronal network of Caenorhabditis elegans." Phil. Trans. R. Soc. B 373.1758 (2018): 20170377.

Author response: 

The revised version of the manuscript addresses all the points raised by the reviewer. We have now added a whole section on whole-brain imaging, which incorporates the papers suggested by the reviewer, including a discussion on whole-brain imaging in immobilized worms. In addition, we now discuss the computational strategies used to address the functional connectome as a separate section.  

Reviewer 2 Report

This paper reviews how new experimental methods will better exploit and detail the working of the neuronal connectivity of the nematode C. elegans. The paper does not read well.  Although I might often be in agreement with what they are trying to say, the message is often lost in weak grammar and phrasing. This is quite disappointing as the topic is timely and of considerable current interest.

 I recommend the authors ask for help from a literate friend.

Figure 2 is trying to do too much, though I’m still unclear about the goals here.  The legend mentions quite a few factors (gene expression levels, sex. peptides) that are not actually included in the diagrams. Graphically, how is the reader supposed to find the “extrasynaptic connections”? Take a step back and show pairings of one circuit at a time, comparing how the male and hermaphrodite conditions differ?  Or how certain circuits differ when gene expression level has been altered? The current version does not communicate well

Author Response

Reviewer 2

Comments and Suggestions for Authors

Reviewer Comment 1: This paper reviews how new experimental methods will better exploit and detail the working of the neuronal connectivity of the nematode C. elegans. The paper does not read well.  Although I might often be in agreement with what they are trying to say, the message is often lost in weak grammar and phrasing. This is quite disappointing as the topic is timely and of considerable current interest.

 I recommend the authors ask for help from a literate friend.

Author response: 

We thank the reviewer for their enthusiasm of the topic being discussed and accept some of the criticism about the grammar and phrasing. The revised version of the manuscript has been carefully and extensively edited for both grammar and phrasing. In addition, the PI has had the revised version read by his colleagues and the manuscript has been suitably edited based on their comments and suggestions.  

Reviewer Comment 2: Figure 2 is trying to do too much, though I’m still unclear about the goals here.  The legend mentions quite a few factors (gene expression levels, sex. peptides) that are not actually included in the diagrams. Graphically, how is the reader supposed to find the “extrasynaptic connections”? Take a step back and show pairings of one circuit at a time, comparing how the male and hermaphrodite conditions differ?  Or how certain circuits differ when gene expression level has been altered? The current version does not communicate well

Author response:

We thank the reviewer for pointing out the overzealousness of Figure 2. Our goal for Figure 2 was to highlight how we can decipher the functional connectome using examples of certain circuits that are already characterized. Upon reviewing Figure 2 and the text associated with the figure, we agree that the figure is attempted to convey too much. The revised version of the manuscript has now broken the figure into two figures. The “revised Figure 2” discusses how distinct signaling mechanisms mediate same avoidance behavior to different stimuli or elicit different behaviors in the two sexes. We have now made a “new Figure 3”, which discusses the role of physiological states in regulating neural circuits by recruiting neuropeptides and neuromodulators. 

Reviewer 3 Report

Overall this is a good review. It is a big topic and difficult to cover everything but I thought they generally chose good examples and discussed them accurately.  I have only a few minor suggestions:

In the abstract (line 11) the authors mention an extrasynaptic connectome, but there is no reference cited.

2. In the discussion of genetically-encoded calcium indicators (line 140), the authors state that this technique requires the use of microfluidics, but there are many cases where this approach has been used in glued or freely-moving animals.

3. The papers cited for microfluidics are also not the ones I would have chosen; maybe it would be good to cite the first use in worm calcium imaging i.e. Chronis et al?

4. In figure 2, the authors use the term "command interneurons", where I think "premotor interneurons" would be more appropriate.

Author Response

Reviewer 3

Comments and Suggestions for Authors

Overall this is a good review. It is a big topic and difficult to cover everything but I thought they generally chose good examples and discussed them accurately.  I have only a few minor suggestions:

Reviewer Comment 1: In the abstract (line 11) the authors mention an extrasynaptic connectome, but there is no reference cited.

Author response: 

We thank the reviewer for their comments on the manuscript. We have revised the abstract and moved the text discussing the extrasynaptic connectome with appropriate references under Introduction (Page 3, Line 58) of the revised manuscript.

Reviewer Comment 2: In the discussion of genetically-encoded calcium indicators (line 140), the authors state that this technique requires the use of microfluidics, but there are many cases where this approach has been used in glued or freely-moving animals.

Author response: 

We thank the reviewer for this comment and have now added complete sections in the manuscript that discussed the use of genetically-encoded calcium sensors in both glued and freely-moving worms.

Reviewer Comment 3: The papers cited for microfluidics are also not the ones I would have chosen; maybe it would be good to cite the first use in worm calcium imaging i.e. Chronis et al?

Author response: 

We agree with the reviewer that first use of calcium imaging in C. eleganswas the Chronis et al. paper and apologize for this oversight. The revised version of the manuscript has a completely rewritten section of calcium imaging with the correct citations. 

Reviewer Comment 4. In figure 2, the authors use the term "command interneurons", where I think "premotor interneurons" would be more appropriate.

Author response: 

We agree with the reviewer’s suggestion on the use of the term ‘premotor interneurons’. All of the revised figures incorporate this suggestion.

Round 2

Reviewer 1 Report

The authors significantly revised the manuscript by reorganizing it and adding new descriptions of  state of the art technologies in whole-brain imaging, modeling and computational functional connectomics approaches. As a result, completeness of the manuscript and description of current research greatly improved. Overall the current version of the review presents a broad and fresh view of the study of connectomics and function in C. elegans and therefore I would recommend its publication in JDB given the authors address the textual inconsistencies and finalize the discussion.

The revision introduced several typos and incoherent sentences, especially in the new Computational Strategies section (some of them listed below). These need to be corrected. Also the novel parts added to the paper were not mentioned in the discussion. Their inclusion is recommended.

Typos and comments:

Abstract:

neural networks modulate *process* sensory information processing

ln 121: 

The first *ver* microfluidic environment used

ln 157: 

to ultimately produce volumetric imaging (60, 61) Both methods

ln 207:

that use known data to predict responses to novel situations (77) C. elegans

ln 208: 

C. elegans are a powerful biophysical *?* with behaviors that can be understood at the level of neural stimulation and robust movement output.

ln 376:

"Analysis of the functional connectome has advanced in the development of in silico models.

Driven by software developments, C. elegans has been modeled by programs like OpenWorm (104, 105).

Discussion of modeling described in computational strategies is missing. 

Author Response

The authors significantly revised the manuscript by reorganizing it and adding new descriptions of state-of-the-art technologies in whole-brain imaging, modeling and computational functional connectomics approaches. As a result, completeness of the manuscript and description of current research greatly improved. Overall the current version of the review presents a broad and fresh view of the study of connectomics and function in C. elegans and therefore I would recommend its publication in JDB given the authors address the textual inconsistencies and finalize the discussion.

Author response: We thank the reviewer for their positive views on our revised manuscript. 

Reviewer Comment: The revision introduced several typos and incoherent sentences, especially in the new Computational Strategies section (some of them listed below). These need to be corrected. Also the novel parts added to the paper were not mentioned in the discussion. Their inclusion is recommended.

Author response: We have now thoroughly proofread the manuscript including the newly added sections for grammatical errors and typos and corrected any remaining mistakes. Also, we have added more text in the discussion about the novel sections added in the revised manuscript.   

Reviewer Comment 

Typos and comments:

Abstract:

neural networks modulate *process* sensory information processing

ln 121:

The first *ver* microfluidic environment used

ln 157:

to ultimately produce volumetric imaging (60, 61) Both methods

ln 207:

that use known data to predict responses to novel situations (77) C. elegans

ln 208:

C. elegans are a powerful biophysical *?* with behaviors that can be

understood at the level of neural stimulation and robust movement output.

ln 376:

"Analysis of the functional connectome has advanced in the development of in

silico models.

Driven by software developments, C. elegans has been modeled by programs like OpenWorm (104, 105).

Author response: We thank the reviewer for pointing out these typos. All the typos mentioned above by the reviewer have been corrected.  In addition, we have proofread the manuscript for any other grammatical errors and edited the text accordingly.

Reviewer Comment: Discussion of modeling described in computational strategies is missing.

Author response: We agree with the reviewer that the discussion lacked the modeling aspects of analyzing the functional connectome. We have added a complete paragraph in the discussion describing the same in the revised version of the manuscript.

Added text:

Whole-brain imaging studies suggest that a population coding mechanism allows for the smooth transitioning of network activity (59-61). This allows the worm to switch between different programs (forward to backward or vice versa) during locomotor behaviors. Novel computational methods will need to be developed to verify in a quantitative manner that population-level features indeed encode behaviors. Moreover, whole-brain imaging in freely moving worms should reveal whether other possible population-level features have indeed behavioral correlates. Optogenetics is especially well suited to uncovering compartment-specific processes. Developing the optogenetic toolkit further to localize photosensitive proteins to specific sub-cellular locations with precise activation is an area of future research. These technologies will help decipher sub-cellular dynamics of sensory and interneurons. 

In silico approaches that utilize the C. elegans connectome to model known behaviors and also predict novel outcomes to a known stimulus are currently being developed. One such open-source platform is the OpenWorm, which aims of building a complete digital organism to simulate all features of C. elegans' behavior(104105). Computational modeling using novel algorithms and superimposition of these models on the experimental data can provide insights into how network/s function after stimulus exposure (74-76). However, developing models that can predict network function based on simulations is still an area that requires further study. Additionally, dynamics of multitudes of neurons during a certain behavior, allow for new approaches in modeling incorporating both the structural connectome data and layering it with the neurophysiological responses and interactions (78). The ‘Dynome’ model depicts the dynamical systems overlaying the structural connectivity (78). These models are more akin to the realistic nervous system and have amazing potential for revealing novel neural pathways and functionalities of the network (6979).

Reviewer 2 Report

This review covers several very interesting topics. The overall readability is much improved, although there are still typos, grammatical errors, and repetitive phrasing to be fixed throughout the document. More generally, sentence structures are often too long and become convoluted. Divide thoughts into shorter sentences, and the ideas will usually become more clear.  As an example, lines 134-136 couple two sentences into one confusing product. This is just an example of a problem running throughout the text.

Lines 18-19, line 121, 188-190, 204-209, and 318-319 are garbled.  Often lacking the verb, noun, or another key sentence part.

Lines 57-67                   Authors seem to misunderstand how “sensory neurons” are defined. The key factor is not their preponderance of synaptic outputs, but their specialized dendritic inputs. Sense cells are not just reflecting inputs from other neurons (as claimed on line 62), but are reacting to inputs from the external environment

Figure 3A.  The graph lumps multiple downstream cells from ASH, and the legend asserts that the different cells eventually end up causing the same final behavior. Legend seems to say that either type of interneuron can independently lead to the same behavioral output. Is it clear that they are not working together to produce that output? Is there a reference to be cited here?

Figure 4.  The legend is not helpful in explaining some of the graphics here.  What is the difference between dotted lines, solid lines, red lines, or blue lines?

Author Response

This review covers several very interesting topics. The overall readability is much improved, although there are still typos, grammatical errors, and repetitive phrasing to be fixed throughout the document. More generally, sentence structures are often too long and become convoluted. Divide thoughts into shorter sentences, and the ideas will usually become moreclear. As an example, lines 134-136 couple two sentences into one confusing product. This is just an example of a problem running throughout the text.

Author response: We thank the reviewer for the positive comments on the readability of the manuscript. Upon checking the manuscript, we did discover some typos and apologize for the oversight. The second revision of the manuscript has been reedited and addresses these typos and improves the grammar used in the manuscript.

Reviewer Comment: Lines 18-19, line 121, 188-190, 204-209, and 318-319 are garbled. Often lacking the verb, noun, or another key sentence part.

Author response: We thank the reviewer for pointing out the grammatical errors. We have rewritten the sentences in question to improve the grammar of the text. In addition, we have 

Reviewer Comment: Lines 57-67 Authors seem to misunderstand how “sensory neurons” are defined. The key factor is not their preponderance of synaptic outputs, but their specialized dendritic inputs. Sense cells are not just reflecting inputs from other neurons (as claimed on line 62), but are reacting to inputs from the external environment.

Author response: The authors recognize that definition of sensory neurons as stated in the text is not just a reflection of the inputs from other neurons but a more primary role in detecting stimuli. This version of manuscript has added additional text to clarify this confusion. 

Original text:

C. elegansneurons can be divided into three functional “classes” of neurons: sensory neurons, motor neurons, and interneurons or premotor neurons (68). The sensory neurons account for a third of the neurons with more connections being pre-synaptic than post-synaptic. Conversely, motor neurons, another third of all neurons, have more post-synaptic connections. The remaining neurons are considered to be interneurons, with large numbers of both pre- and post-synaptic connections (9). Having more pre-synaptic connections, sensory neurons are able to gather more signals from surrounding neurons. Post-synaptic connections within motor neurons allows them more synaptic connections with muscles, thereby producing fine-tuned actions. Understanding the connections alone between these neuron classes helps increase our understanding of how a signal is transduced, processed, to ultimately produce behavioral outputs, as seen in previous work on C. elegans’ navigation (14). However, these actions cannot be understood by the static connections alone. 

Modified text:

C. elegansneurons can be divided into three functional “classes” of neurons: sensory neurons, motor neurons, and interneurons or premotor neurons (68). The sensory neurons have dendrites that extend to the tip of the nose and terminate into diverse ciliated structures to detect stimuli from the environment. These neurons account for a third of the neurons with more connections being pre-synaptic than post-synaptic. Conversely, motor neurons, another third of all neurons, have more post-synaptic connections. The remaining neurons are considered to be premotor interneurons, with large numbers of both pre- and post-synaptic connections (9). Understanding the connections alone between these neuron classes helps increase our understanding of how a signal is transduced, processed, to ultimately produce behavioral outputs, as seen in previous work on C. elegans’ navigation (14).

Reviewer Comment: Figure 3A. The graph lumps multiple downstream cells from ASH, and the legend asserts that the different cells eventually end up causing the same final behavior. Legend seems to say that either type of interneuron can independently lead to the same behavioral output. Is it clear that they are not working together to produce that output? Is there a reference to be cited here?

Author response: We would like to clarify this query by referring to results from multiple papers (Hart et al., 1995; Maricq et al., 1995; Mellem et al., 2002, Hilliard et al., 2005).Differential activation of ASH results in differential release of neurotransmitter and distinct patterns of downstream signaling (Mellem et al., 2002). This differential release of glutamate from ASH activates different types of glutamate receptors on postsynaptic premotor interneurons. Weak activation, such as that elicited by nose touch, activates non-NMDA ionotropic glutamate receptor (iGluR) subunits GLR-1 and/or GLR-2 (Hart et al., 1995; Maricq et al., 1995; Mellem et al., 2002). Whereas stimuli that evoke higher levels of Ca2+ release, such as hyperosmolarity (Hilliard et al., 2005), can activate not only GLR-1/ GLR-2 channels but also NMR-1 and NMR-2-containing NMDA iGluR (Mellem et al., 2002). Hence, the intensity and type of aversive stimuli are decoded through differential activation of postsynaptic glutamate receptors in the premotor interneurons. We hope that this clarifies the doubts that the reviewer raised.

We have now edited and cited the appropriate references in the figure legend. 

Original:

Figure 3. Differential use of neurons and pathways to create functional circuits. A) The same neuron, ASH, responds to two different stimuli, using different intracellular signaling pathways and post synaptic targets to give rise to the same behavioral avoidance output. Osmotic stress is sensed and signaled via osm-10 and targets the NMDA-type receptor nmr-1 whereas nose touch utilizes itr-1 and the glutamate receptor glr-1.

Modified:

Functional circuits are created by differential use of neurons and distinct intra- and inter-cellular signaling pathways. A) The sensory neuron ASH responds to two different stimuli resulting in an avoidance response via the premotor interneurons AVA, AVB, AVD & AVE. Osmotic stress is mediated via osm-10whereas nose touch utilizes itr-1within ASH. Downstream of ASH, osmotic stress targets the NMDA-type receptor nmr-1, whereas nose touch activates the glutamate receptor glr-1in the premotor interneurons(86,88).

Reviewer Comment:Figure 4. The legend is not helpful in explaining some of the graphics here. What is the difference between dotted lines, solid lines, red lines, or blue lines?

Author response: Upon reassessment of the figure, we concluded that the figure was not helping convey the major point of the effects of different behavioral states in shaping functional circuits. Using the example of roaming and dwelling behavioral states, we have redrawn the figure (New Figure 4) to explain the role of multiple neurons in modulating the circuit in a connectome independent manner. We show that a distributed circuit modulates the transitions between these two states. We have also rewritten the legend to clarify and explain different aspects of the figure.  
